# Quantifying the effect of intervertebral cartilage on neutral posture in the necks of sauropod dinosaurs

Michael P. Taylor

Department of Earth Sciences, University of Bristol, Bristol, England, United Kingdom

## ABSTRACT

Attempts to reconstruct the neutral neck posture of sauropod dinosaurs, or indeed any tetrapod, are doomed to failure when based only on the geometry of the bony cervical vertebrae. The thickness of the articular cartilage between the centra of adjacent vertebrae affects posture. It extends (raises) the neck by an amount roughly proportional to the thickness of the cartilage. It is possible to quantify the angle of extension at an intervertebral joint: it is roughly equal, in radians, to the cartilage thickness divided by the height of the zygapophyseal facets over the centre of rotation. Applying this formula to published measurements of well-known sauropod specimens suggests that if the thickness of cartilage were equal to 4.5%, 10% or 18% of centrum length, the neutral pose of the *Apatosaurus louisae* holotype CM 3018 would be extended by an average of 5.5, 11.8 or 21.2 degrees, respectively, at each intervertebral joint. For the *Diplodocus carnegii* holotype CM 84, the corresponding angles of additional extension are even greater: 8.4, 18.6 or 33.3 degrees. The cartilaginous neutral postures (CNPs) calculated for 10% cartilage—the most reasonable estimate—appear outlandish. But it must be remembered that these would not have been the habitual life postures, because tetrapods habitually extend the base of their neck and flex the anterior part, yielding the distinctive S-curve most easily seen in birds.

## INTRODUCTION

The habitual posture of the necks of sauropod dinosaurs has been controversial ever since their body shape has been understood. Both elevated and more horizontal postures have been depicted, sometimes even in the same images—for example, *Knight*'s classic *1897* painting of *Apatosaurus* and *Diplodocus* (Fig. 1). See the introduction to *Taylor & Wedel (2013)* for a more comprehensive historical overview.

*Stevens & Parrish (1999)* used DinoMorph, a computer program of their own devising, to model the intervertebral articulations in the necks of two well-known sauropods, *Apatosaurus* and *Diplodocus*. They found that when the vertebrae were best aligned—with the centra in articulation and the zygapophyseal facets maximally overlapped—the necks were held in roughly horizontal positions; *Stevens & Parrish (1999)* concluded without further discussion that this was the habitual posture in life—an assumption which they

Corresponding author
Michael P. Taylor,
dino@miketaylor.org.uk

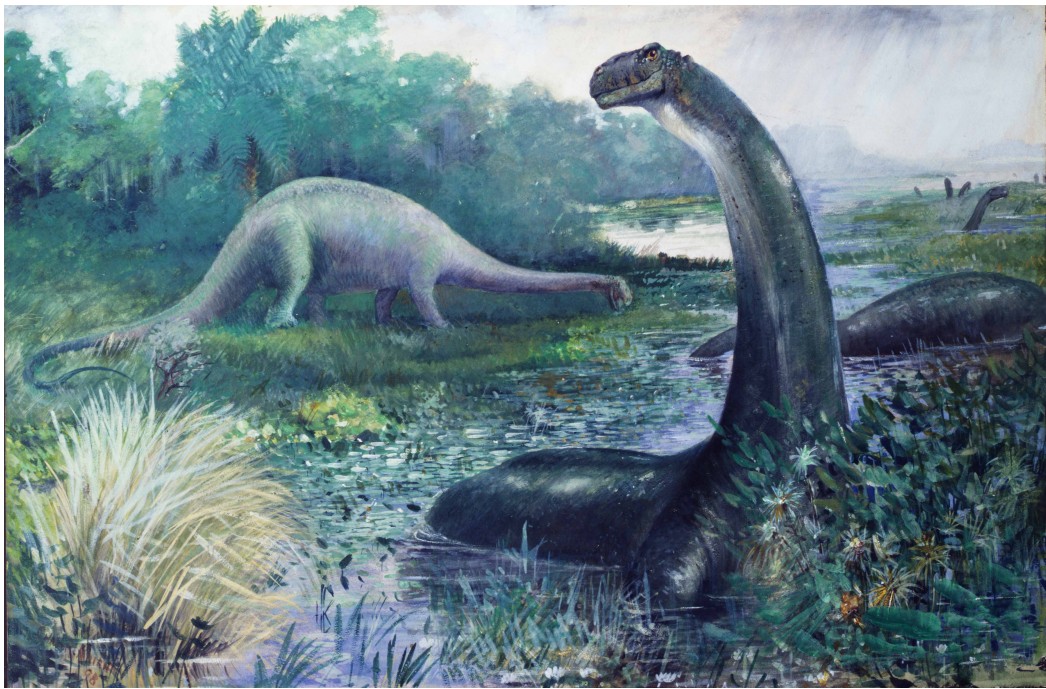

**Figure 1 Charles R. Knight's famous 1897 painting of sauropods, which were then considered amphibious.** In the foreground, *Apatosaurus* ("*Brontosaurus*" of his usage) wades in a lake, its neck erect. In the background, *Diplodocus* wanders on the shore, its neck held low and horizontal. These differences in posture may not represent different perceptions of the habitual behaviour of these different taxa, merely the postures these individuals happened to adopt at a particular moment.

subsequently asserted was supported by observation of extant tetrapods (*Stevens & Parrish, 2005*). In fact, as discussed below, tetrapods do not habitually hold their necks in neutral pose; nevertheless, determining neutral pose is an important step towards understanding habitual pose.

The study of *Stevens & Parrish (1999)* has been influential but can and should be further refined. *Taylor & Wedel (2013)* demonstrated the important role of a neglected element, the intervertebral cartilage that separates the centra of adjacent vertebrae. We noted in that paper that including the cartilage in models affects the neutral posture recovered, causing the neck to be raised more than when only bone is taken into account, but we failed to quantify the additional extension of the neck. The present paper remedies this deficiency.

The neutral pose determined by Stevens and Parrish from bones alone is termed osteological neutral pose (ONP). I use the term cartilaginous neutral pose (CNP) for the pose found when intervertebral cartilage is included. Each specimen has a true CNP, determined by the actual arrangement of cartilage on its vertebrae. Because we are dealing with extinct animals known only from fossils, we must make assumptions about the cartilage that existed in life, and so can derive only provisional CNPs.

Note that zygapophyseal cartilage has no or negligible effect on the angle of extension between vertebrae. This is partly because this cartilage is so thin compared with that between consecutive centra, but primarily because of the orientation of the zygapophyseal

facets. If they faced anteriorly and posteriorly, then inserting cartilage between them would push the dorsal part of the vertebral articulation apart and deflect the neutral pose downwards. But because the facets face dorsomedially and ventrolaterally, the addition of cartilage between them does not affect their relative anteroposterior position.

## METHODS

### Formula for additional extension

Figure 2A shows two adjacent vertebrae in osteological neutral pose (ONP): the condyle (anterior ball) of one vertebra is nestled in the cotyle (posterior cup) of the other, and its prezygapophyseal facets are maximally overlapped with the postzygapophyseal facets of the other.

Figure 2B shows the effect of including intervertebral cartilage of thickness $t$ (here depicted as being one tenth as thick as the length of the bony centrum). The cartilage itself is shown in black. For simplicity, it is depicted as though all are attached to the condyle of the more posterior (grey) vertebra; in fact it would have been roughly half and half on this condyle and on the cotyle of the more anterior (blue) vertebra.

In order to accommodate the intervertebral cartilage, the cotyle of the anterior vertebra has to be shifted forward by a distance equal to the thickness of the cartilage, as shown in Fig. 2B. But in this new "neutral pose", the zygapophyseal facets remain maximally overlapped, so the effect is to rotate the anterior vertebra anti-clockwise about the centre of the zygapophyses, which is at height $h$ above the midline of the condyle. The red lines are drawn between the centre of rotation and the anteriormost point of the bony condyle and the cartilage extension (or, equivalently, the deepest part of the cotyles of both the yellow and blue vertebrae). The rotation between the blue and yellow vertebrae is equal to the angle $\theta$ between the red lines.

Because the thickness of cartilage is a small proportion of centrum length, this angle is small. Therefore a line drawn from the anteriormost point of the bony centrum to that of the cartilage (short line of length $t$ in Fig. 3) forms a triangle with the red lines that is close to a right-angled triangle. Consider the angle $\theta$: its opposite is the short line of length $t$ and its hypotenuse is one of the long lines of length $h$. Therefore $\sin(\theta) = t/h$. But for small angles, $\sin(\theta) \approx \theta$ (measured in radians).

Therefore, the angle of extension due to cartilage at an intervertebral joint, in radians, is approximately equal to the thickness of the cartilage divided by the height of the zygapophyses above half height of the joint between centra.

$$\theta = t/h.$$

This formula is independent of the unit of linear measurement: inches, millimetres or pixels in a digitised photograph are all equally valid so long as the same unit is used for cartilage thickness and zygapophyseal height.

Since $\pi$ radians is equal to 180° (half a circle), an angle in radians can be converted to degrees by multiplying by $180/\pi$. Therefore, the angle of extension in degrees is $t/h \times 180/\pi$.

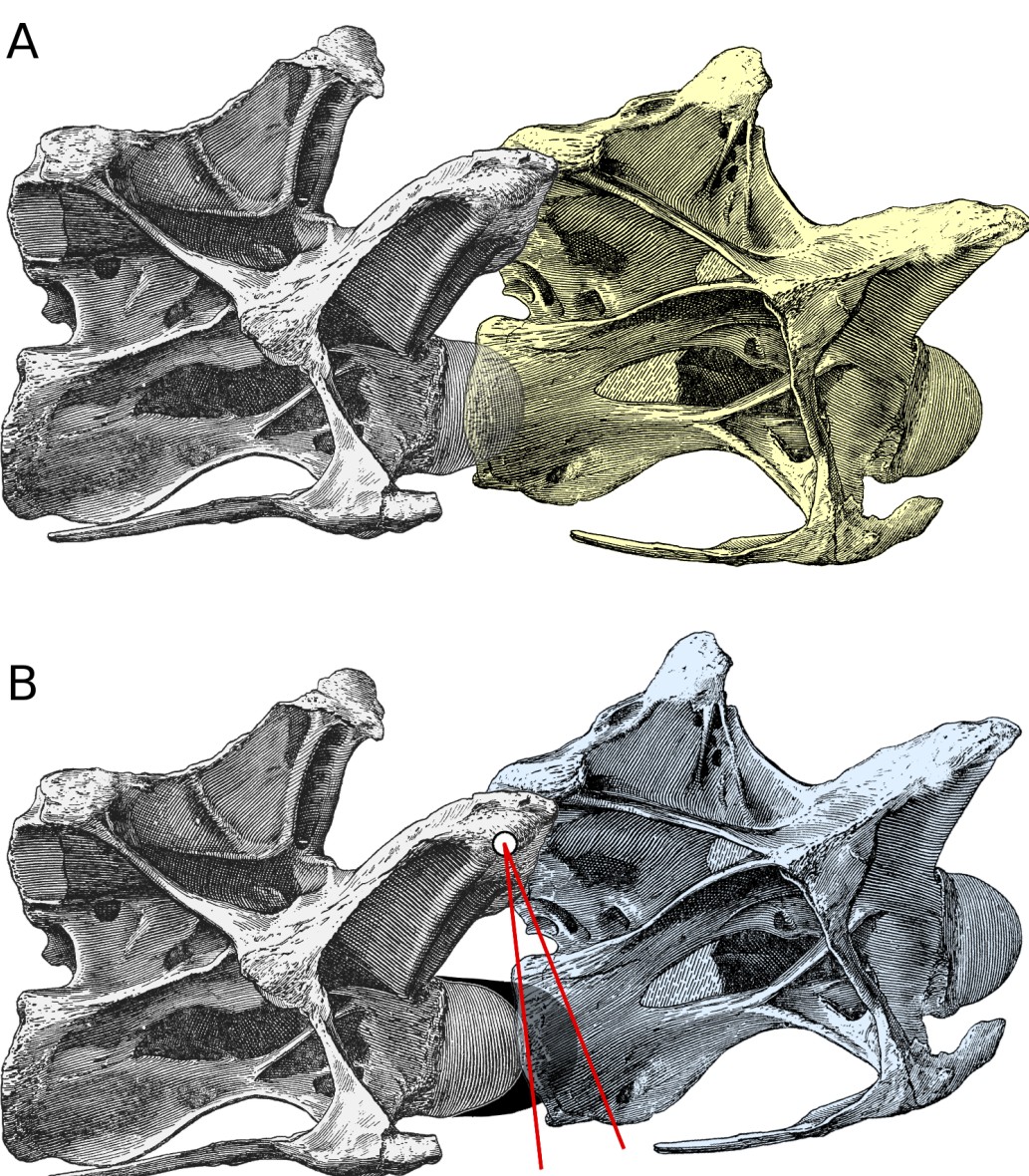

**Figure 2** **Increased angle of elevation at an intervertebral joint when cartilage is included. Posterior cervical vertebrae 13 and 14 of *Diplodocus carnegii* holotype CM 84, from *Hatcher* (*1901*: plate III), in right lateral view.** (A) C13 (yellow) in osteological neutral posture, with the condyle of C14 embedded in its cotyle and with zygapophyseal facets maximally overlapped. (B) Intervertebral cartilage (black) added, and C13 (blue) rotated upwards to accommodate it. (For simplicity, the cartilage is depicted as though all attached to the condyle of the posterior vertebra in the present figure and in Fig. 3; in fact it would have been roughly half and half on this condyle and on the cotyle of the more anterior vertebra.) Since the zygapophyses remain maximally overlapped, the centre of their facets forms the axis of rotation (white dot); red lines join the centre of rotation to the most anterior point of the bony condyle and of the intervertebral cartilage. By similarity, the angle between the yellow and blue vertebrae is equal to that between the red lines.

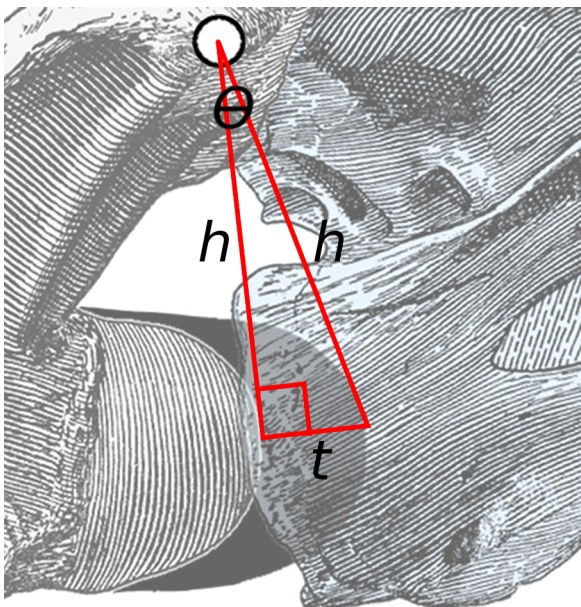

**Figure 3  Close-up of area of rotation in Fig. 2.** The two long lines, each of length *h*, connect the middle of the zygapophyseal facets to the anteriormost point of the condyle of the posterior vertebra and the cotyle of the anterior one. The short line of length *t* is projected at a right angle to the left line, and more or less connects the points on the condyle and cotyle. The angle between the two long lines is $\theta$.

This calculation is only approximate: the triangle is not a true right-triangle, and $\sin(\theta)$ is only approximately equal to $\theta$. However, these minor sources of inaccuracy are dwarfed by other sources of error when working with sauropods: distortions in the measured vertebrae, estimations in measurement where the vertebrae are incomplete, and uncertainty about cartilage thickness. In this context, the $\theta = t/h$ approximation is quite precise enough.

## Cartilage thickness assumptions

*Taylor & Wedel* (*2013*: 7–8) recently estimated the thickness of intervertebral cartilage from vertebral spacing between adjacent vertebrae in two sauropod genera. We found that cartilage thickness between cervical vertebrae of an adult *Sauroposeidon* individual was about 4.5% of centrum length; that between anterior dorsal vertebrae of a subadult *Apatosaurus* individual CM 3390 it was about 20% of centrum length; and that between mid-to-posterior dorsal vertebrae of a second, juvenile, *Apatosaurus* individual CM 11339 it was about 15% of centrum length. Assuming similar absolute thickness of cartilage in the neck of adult *Apatosaurus* as in *Sauroposeidon* (about 52 mm), we estimated that cartilage thickness would be about 9.8% the length of the shorter *Apatosaurus* vertebrae. Similarly, assuming similar absolute thickness of cartilage in adult *Apatosaurus* necks as in subadult anterior torsos, we estimated cartilage thickness in adult *Apatosaurus* might have been about 11%, a value fairly consistent with that derived from *Sauroposeidon* measurements.

These cartilage thickness proportions are provisional—we are very aware that our sample is tiny, and encourage other sauropod workers to CT-scan articulated sequences of

vertebrae when possible. However, since they are the only existing estimates, I calculated the effect of inserting intervertebral cartilage into the neck of *Apatosaurus* using three possible thicknesses: the 4.5% of the adult *Sauroposeidon* neck; the 10% that was estimated in two ways as most likely for the adult *Apatosaurus* neck; and 18%, the average of the 20% and 15% found for the two non-adult *Apatosaurus* torso sequences. Since *Diplodocus* is closely related to *Apatosaurus* and was also discussed by *Stevens & Parrish (1999)*, I also calculated the effect of adding cartilage to its neck in the same proportions as for *Apatosaurus*.

### Sauropod specimens

I used the same well-known specimens as *Stevens & Parrish (1999)*: CM 3018, the holotype of *Apatosaurus louisae*; and CM 84, the holotype of *Diplodocus carnegii*. Both specimens reside in the Carnegie Museum of Natural History, Pittsburgh, Pennsylvania, USA. They are well-preserved for sauropods, having nearly complete cervical sequences, although the more posterior vertebrae of CM 3018 are badly damaged and all the vertebrae suffer from some distortion. All calculations are based only on centrum length, zygapophyseal height (measured from published illustrations) and hypothetical cartilage thicknesses.

## RESULTS

For *Apatosaurus* CM 3018, the results are as shown in Table 1: additional extension across all 13 analysed intervertebral joints sums to 70°, 155° and 279° for 4.5%, 10% and 18% cartilage thickness. Figure 4 shows the effect of the additional extension caused by 10% cartilage compared to a horizontal neck: if osteological neutral pose were horizontal, then the neutral pose when taking into account intervertebral cartilage whose thickness is 10% of centrum length would be as depicted. I term this the 10% cartilaginous neutral pose or 10% CNP. (In fact, *Stevens & Parrish (1999)* found ONP in both *Apatosaurus* and *Diplodocus* to be somewhat below horizontal, but since their exact angles of flexion were not published, it is not possible to determine how their favoured pose would appear when modified by the addition of cartilage.)

For *Diplodocus* CM 84, the results are as shown in Table 2: additional extension across all 13 analysed intervertebral joints sums to 108°, 241° and 434° for 4.5%, 10% and 18% cartilage thickness. Figure 5 shows the effect of the additional extension caused by 10% cartilage compared to a horizontal *Diplodocus* neck, as Fig. 4 does for *Apatosaurus*; the same caveats apply.

## DISCUSSION

The additional angles of extension calculated here are greater for *Diplodocus* than for *Apatosaurus*—on average, about 55% greater. This is for two reasons. First, the additional angle of extension is directly proportional to cartilage thickness, which I calculated as proportional to centrum length, and the centra are longer in *Diplodocus*; and second, the angle is also inversely proportional to the height of the zygapophyseal facets above the centre of rotation between adjacent centra, and this is lower in *Diplodocus*.

**Table 1 Centrum length, zygapophyseal height, possible cartilage thicknesses and corresponding additional angles of extension in the neck of the *Apatosaurus louisae* holotype CM 3018.** Centrum lengths are taken from *Gilmore* (*1936*: 196) except for C5, C14 and C15, which are omitted from Gilmore's table and were instead measured from his illustration (*Gilmore, 1936*: plate XXIV). Zygapophyseal height was measured from the midline of the centrum to the midpoint of the postzygapophysis on plate XXIV. Cartilage thicknesses were calculated as percentages of the centrum lengths, using three different percentages as described in the text. Additional angles of extension were calculated using the formula in the Methods section. Cumulative angles measure the total additional extension from ONP, beginning with small extensions at the shoulder and increasing anteriorly. The full spreadsheet from which this table was exported, including formulae, is Supplemental Information 1.

| Cv# | Centrum length (mm) | Zygapophysis height (mm) | Cartilage (mm) | | | Angle (degrees) | | | Cumulative angle (degrees) | | |
|---|---|---|---|---|---|---|---|---|---|---|---|
| | | | 4.5% | 10% | 18% | 4.5% | 10% | 18% | 4.5% | 10% | 18% |
| 1 | 45 | | 2 | 5 | 8 | | | | | | |
| 2 | 190 | | 9 | 19 | 34 | | | | | | |
| 3 | 280 | 130 | 13 | 28 | 50 | 6 | 12 | 22 | 70 | 155 | 279 |
| 4 | 370 | 150 | 17 | 37 | 67 | 6 | 14 | 25 | 64 | 143 | 257 |
| 5 | 443 | 160 | 20 | 44 | 80 | 7 | 16 | 29 | 58 | 129 | 231 |
| 6 | 440 | 171 | 20 | 44 | 79 | 7 | 15 | 26 | 51 | 113 | 203 |
| 7 | 450 | 155 | 20 | 45 | 81 | 8 | 17 | 30 | 44 | 98 | 176 |
| 8 | 485 | 206 | 22 | 49 | 87 | 6 | 13 | 24 | 37 | 81 | 146 |
| 9 | 510 | 285 | 23 | 51 | 92 | 5 | 10 | 18 | 30 | 68 | 122 |
| 10 | 530 | 273 | 24 | 53 | 95 | 5 | 11 | 20 | 26 | 57 | 103 |
| 11 | 550 | 308 | 25 | 55 | 99 | 5 | 10 | 18 | 21 | 46 | 83 |
| 12 | 490 | 261 | 22 | 49 | 88 | 5 | 11 | 19 | 16 | 36 | 65 |
| 13 | 480 | 290 | 22 | 48 | 86 | 4 | 9 | 17 | 11 | 25 | 46 |
| 14 | 411 | 274 | 19 | 41 | 74 | 4 | 9 | 15 | 7 | 16 | 29 |
| 15 | 372 | 292 | 17 | 37 | 67 | 3 | 7 | 13 | 3 | 7 | 13 |
| | | **Average** | 18.3 | 40.3 | 72.5 | 5.5 | 11.8 | 21.2 | | | |

There is no denying that the cartilaginous neutral poses (CNPs) described here for *Apatosaurus* and *Diplodocus* appear outlandish. Using the largest of the candidate cartilage thicknesses, 18% of centrum length, the neutral pose for *Diplodocus* has C3 oriented at 434° to the horizontal (Table 2, last column)—that is, the neck would be extended all the way around through 360° and a further 74°. This alone seems to be enough to discount the possibility that the 18% estimate of cartilage thickness is correct—not unreasonably, since this was measured from the dorsal sequences of sub-adult and juvenile specimens. However, the 10% cartilage thickness that seems the best estimate also yields surprising neutral postures (Figs. 4 and 5). It is tempting for this reason to prefer the 4.5% cartilage thickness, which results in C3 of *Diplodocus* extending only 108°—although note that even this is well past vertical. However, it seems unlikely (based on our small sample of CT scans) that half-meter-long *Apatosaurus* cervicals can have been separated by as little as 23 mm of cartilage. At present, 10% of centrum length is our best estimate of cartilage thickness.

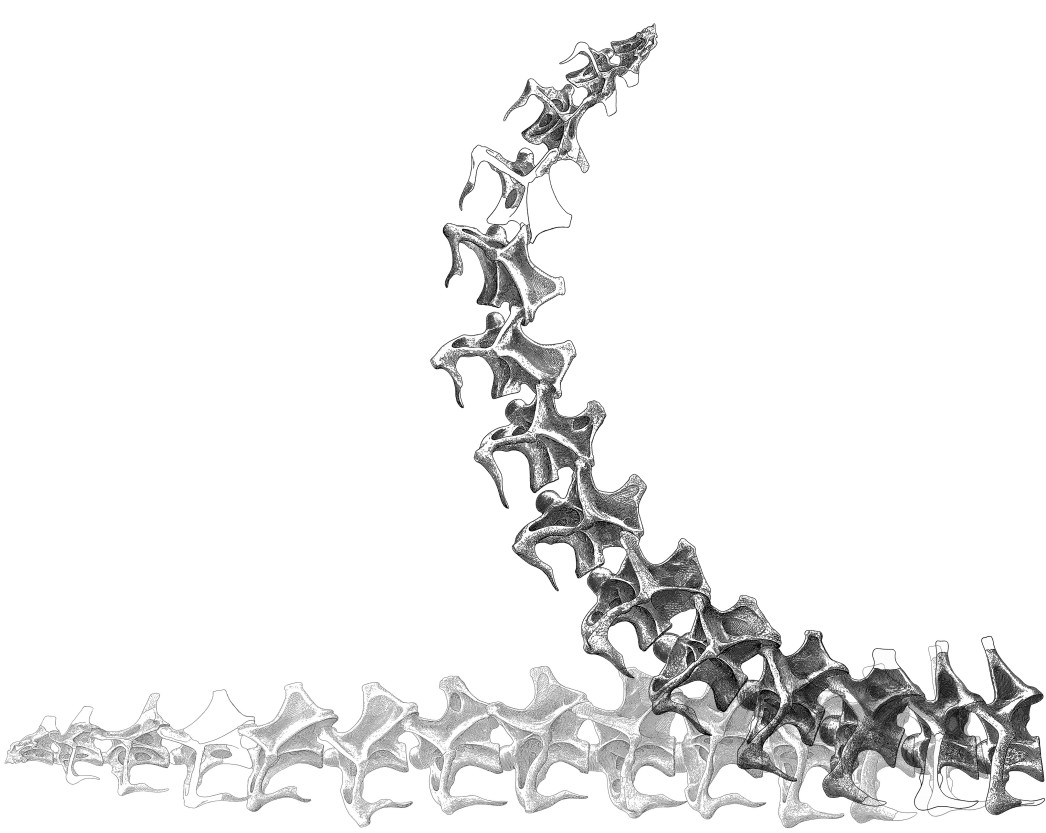

**Figure 4 Effect of adding cartilage to the neutral pose of the neck of *Apatosaurus louisae* CM 3018.** Images of vertebrae from *Gilmore* (*1936*: plate XXIV). At the bottom, the vertebrae are composed in a horizontal posture. Superimposed, the same vertebrae are shown inclined by the additional extension angles indicated in Table 1. If the slightly sub-horizontal osteological neutral pose of *Stevens & Parrish (1999)* is correct, then the cartilaginous neutral pose would be correspondingly slightly lower than depicted here, but still much closer to the elevated posture than to horizontal. (Note that the posture shown here would *not* have been the habitual posture in life: see discussion).

The CNP for other dinosaurs may be even more extreme than for sauropods. *Samman (2013)* articulated the cervical series of *Tyrannosaurus* (using a composite of two specimens, FMNH PR 2081 and TCMI 2001.90.1). She found that the centra alone articulate naturally into an 'S'-curve, due to their keystoned shapes in lateral view, but that when the zygapophyses are also articulated, the ONP was strongly extended into a posture that would surely not have been adopted in life. Inserting articular cartilage between the centra would raise this posture yet further.

Although the 10% CNP calculated and illustrated in this paper is a more defensible neutral pose than the ONP of *Stevens & Parrish (1999)*, I must emphasised that I do *not* suggest this was the habitual pose in life. As noted by *Vidal, Graf & Berthoz (1986)* and *Taylor, Wedel & Naish (2009)*, live tetrapods do not habitually hold their necks in neutral pose. Instead, when awake and alert, they extend (raise) the base of the neck and flex (lower) the anterior part. The result is that the middle part of the cervical column is habitually held much more vertically in most tetrapods that would be apparent from the

**Table 2 Centrum length, zygapophyseal height, possible cartilage thicknesses and corresponding additional angles of extension in the neck of the *Diplodocus carnegii* holotype CM 84.** Centrum lengths are taken from *Hatcher* (*1901*: 38). Zygapophyseal height was measured from the midline of the centrum to the midpoint of the postzygapophysis on *Hatcher* (*1901*: plate III). Cartilage thicknesses, angles and cumulative angles are as for Table 1. The full spreadsheet from which this table was exported, including formulae, is Supplemental Information 2.

| Cv# | Centrum length (mm) | Zygapophysis height (mm) | Cartilage (mm) | | | Angle (degrees) | | | Cumulative angle (degrees) | | |
|---|---|---|---|---|---|---|---|---|---|---|---|
| | | | 4.5% | 10% | 18% | 4.5% | 10% | 18% | 4.5% | 10% | 18% |
| 1 | | | | | | | | | | | |
| 2 | 165 | | 7 | 17 | 30 | | | | | | |
| 3 | 243 | 64 | 11 | 24 | 44 | 10 | 22 | 39 | 108 | 241 | 434 |
| 4 | 289 | 59 | 13 | 29 | 52 | 13 | 28 | 50 | 99 | 219 | 395 |
| 5 | 372 | 108 | 17 | 37 | 67 | 9 | 20 | 35 | 86 | 192 | 345 |
| 6 | 442 | 132 | 20 | 44 | 80 | 9 | 19 | 34 | 77 | 172 | 309 |
| 7 | 485 | 108 | 22 | 49 | 87 | 12 | 26 | 46 | 69 | 153 | 275 |
| 8 | 512 | 161 | 23 | 51 | 92 | 8 | 18 | 33 | 57 | 127 | 229 |
| 9 | 525 | 161 | 24 | 53 | 95 | 8 | 19 | 34 | 49 | 109 | 196 |
| 10 | 595 | 209 | 27 | 60 | 107 | 7 | 16 | 29 | 41 | 90 | 162 |
| 11 | 605 | 202 | 27 | 61 | 109 | 8 | 17 | 31 | 33 | 74 | 133 |
| 12 | 627 | 233 | 28 | 63 | 113 | 7 | 15 | 28 | 25 | 57 | 102 |
| 13 | 688 | 239 | 31 | 69 | 124 | 7 | 17 | 30 | 18 | 41 | 74 |
| 14 | 642 | 271 | 29 | 64 | 116 | 6 | 14 | 24 | 11 | 25 | 44 |
| 15 | 595 | 309 | 27 | 60 | 107 | 5 | 11 | 20 | 5 | 11 | 20 |
| | | **Average** | 21.9 | 48.6 | 87.4 | 8.4 | 18.6 | 33.3 | | | |

fleshy envelope (*Wedel & Taylor, 2014*). Indeed, in many mammals that we hardly even think of as having a neck, the vertebral column bends backwards beyond the vertical: this is seen for example in rabbits, mice and guinea pigs as well as cats and chickens (*Vidal, Graf & Berthoz, 1986*: figs. 2–5, 7, 8). Accordingly, we would expect that the life poses of sauropods had the base of the neck extended yet further than the angles here shown as neutral; but that the anterior part of their necks would have been curved forwards and downwards. It seems possible that in both diplodocids analysed here, part of the neck habitually curved backwards beyond the vertical in an "S" shape, as in many extant birds.

Similarly, *Tyrannosaurus* must have habitually held its neck in a pose differing greatly from its neutral posture. In particular, much of its neck must have been flexed downwards most of the time, perhaps extending only when tearing meat from a carcass.

The effect of intervertebral cartilage on neck flexibility, as opposed to its effect on neutral posture, remains to be determined. *Taylor & Wedel* (*2013*: 15) showed that in turkeys, zygapophyseal surfaces are extended by cartilage, and it is likely that this is true of all tetrapods. Larger zygapophyseal facets translate to more flexibility, as a greater displacement from the neutral pose can occur before the facets become disarticulated.

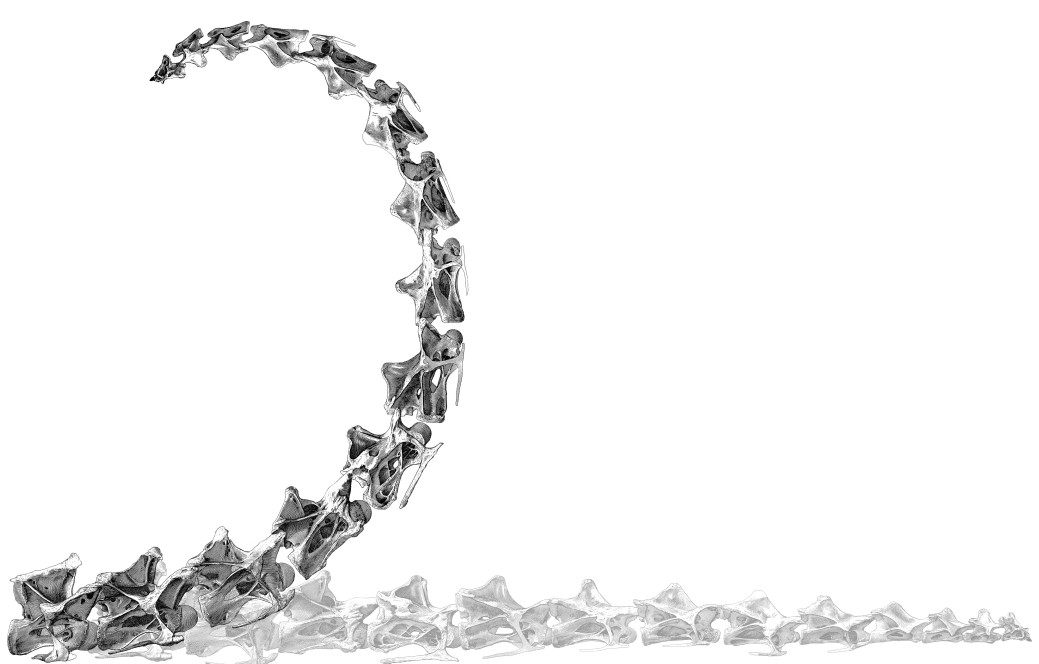

**Figure 5** **Effect of adding cartilage to the neutral pose of the neck of *Diplodocus carnegii* CM 84.** Images of vertebrae from *Hatcher* (*1901*: plate III). At the bottom, the vertebrae are composed in a horizontal posture. Superimposed, the same vertebrae are shown inclined by the additional extension angles indicated in Table 2.

But this is only a relatively small effect (increasing flexibility by about 11% in our turkey specimen) and relates to zygapophyseal rather than intervertebral cartilage.

As noted by *Taylor & Wedel* (*2013*: 15), *Cobley, Rayfield & Barrett (2013)* found that ostrich necks with their soft tissue in place are *less* flexible than bones alone indicate. However, we know that human necks are much more flexible in life than the bones alone would suggest, since the flat articular surfaces of human cervical centra taken alone would indicate an almost entirely inflexible neck. The different effect on neck flexibility of intervertebral cartilage across different taxa would be a fruitful area for further study.

# ACKNOWLEDGEMENTS

This paper could not have been written without numerous stimulating discussions with Matt Wedel (Western University of Health Sciences). I thank Chris Noto (University of Wisconsin Parkside) for his swift, efficient and fair editorial handling of this manuscript. All three reviewers provided careful, detailed, actionable reviews that materially improved this paper: I thank Matt Bonnan (Richard Stockton College of New Jersey), Heinrich Mallison (Museum für Naturkunde Berlin) and Eric Snively (University of Wisconsin La Crosse) for their contributions.

### Funding

The author declares there was no funding for this work.

## Competing Interests

The author declares there are no competing interests.

## Author Contributions

- Michael P. Taylor conceived and designed the experiments, performed the experiments, analyzed the data, wrote the paper, prepared figures and tables, reviewed drafts of the paper.

## Supplemental Information

Supplemental information for this article can be found online at http://dx.doi.org/ 10.7717/peerj.712#supplemental-information.

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
