# Peer review of "Quantifying the effect of intervertebral cartilage on neutral posture in the necks of sauropod dinosaurs"

_PeerJ, doi:10.7717/peerj.712_

## Round 0.1 · original submission · Minor Revisions

The paper overall is strong and I thought best for it to undergo only minor revisions. There is very little that needs to be changed, mostly structural, and it is mostly centered in the introduction and methods. Some rearrangement of sentences in the results and discussion may help with interpretation. I found the suggestions of reviewers 2 and 3 to be particularly helpful, and encourage you to consider them carefully.

I agree with much of what all reviewers noted about the negative or argumentative tone at the beginning of the manuscript and strongly urge you to make changes to address this (multiple reviewer comments on the annotated manuscripts). Also, I think the introduction would be stronger if you did provide some more context to this issue to tie it to your previous paper, though a thorough retelling of the background (like reviewer 1 suggests) is not needed. Though much of the harsher criticisms of reviewer 1 may be dismissed, some clarification in your methods is needed, especially in the description of measurement methods and addressing the number of specimens used in the study, in addition to adding in reference to other pertinent work (see reviewer 3). The methods section would be improved greatly if you included an overall equation for reference. This is also strongly encouraged. These changes will make for a much stronger paper.

Because there were some major differences of opinion on this paper check for inconsistent comments in annotated manuscript copies, and to go with your preferred suggestions. If you have any questions during the revision process, please do not hesitate to contact me.

·

Basic reporting

The article’s writing does not “include sufficient introduction and background to demonstrate how the work fits into the broader field of knowledge.” In fact, the article reads as if it is a note of clarification on the Taylor and Wedel (2013) paper rather than a stand-alone unit. For example, whereas I appreciate that the author is trying to avoid repeating in its entirety the historical perspective of this previous work, a summary of the main hypotheses / take-home points would have been valuable – not everyone reading this article will be familiar with the history of this particular “controversy” in dinosaur paleontology.

Moreover, the article’s tone and language drifts from colloquial to technical and back again, and at times makes statements that could be construed as walking a fine line of “professional standards of courtesy and expression.” For example, whether intentional or not, the sentence in the introduction that states “Stevens and Parrish (1999) concluded without further justification that this was the habitual posture in life” in the article suggests Stevens and Parrish were sloppy or not careful. Nothing could be further from reality. They were the first to quantify sauropod neck posture using computer modeling – at the time, reconstruction of cartilage was not much of a consideration, especially because it was difficult to model. It was certainly a significant contribution in that it spurred the current discussion and investigation of sauropod cervical mobility. By the third paragraph of the very short introduction, the author describes a previous omission by Taylor and Wedel (2013) as “stupid” and claims the paper will correct this “deficit.” Use harsh colloquialisms does nothing to help the paper. Whether intentional or not, such language comes across as unprofessional and perhaps arrogant – even though I realize Taylor is referring to his own work. It is strongly recommended these colloquialisms be removed from the text.

Regarding anatomical terminology, the author is inconsistent. Anatomical terms such as anterior are combined in the same sentences as “backwards.” It is important for ease of understanding and to avoid future confusion that Taylor use consistent anatomical nomenclature.

Experimental design

There is very little experimental design. Essentially, Taylor formulates an angle of intervertebral extension based on data from Taylor and Wedel (2013), and then uses this with two-dimensional figures to establish a Cartilage Neutral Posture (CNP). “The submission should clearly define the research question, which must be relevant and meaningful.” There is not a clear research question, beyond reporting what happens when you add the newly-formulated intervertebral extension data. Major questions arise. Do we know that the angle of extension due to cartilage is going to follow the same pattern at each part of the cervical series? Does this apply to what was found in the turkey from Taylor and Wedel (2013)? In other words, can you show in a modern dinosaur with all its cartilage intact that this angle is more or less constant across the cervical series? None of this is clearly explained. Later in the discussion, Taylor admits that his reconstructions of sauropod “neutral” posture are “outlandish.” One could make the argument that because these data are apparently based on two-dimensional shapes and do not consider the three-dimensional shape of the centra and zygapophyses, you might very well get the “outlandish” ONPs. Can Taylor provide assurances to this effect; e.g., how the two-dimension drawings of vertebrae from Hatcher and Gilmore are standardized?

What I also don’t understand is why in the time since the Taylor & Wedel (2013) publication it has not been possible to at least obtain more turkey necks? These cartilage estimates are apparently (and correct me if I’m wrong) on a single turkey specimen. If you had at least a good sample of just the turkey necks, some of the sample issues would get better. I completely understand that ostrich, camel, etc. necks don’t come along every day, but certainly turkeys, chickens, and the like do. Why not measure the variation of cartilage thickness across several adults in those, and make sure the intervertebral thickness and extension you report are based on a larger sample?

In essence, one is left with the impression that the investigation was not conducted rigorously and to a high technical standard. Perhaps it was, but there is very little to go on in this very short paper to assure that this is indeed the case. I am not suggesting that the data are “good” or “bad;” rather, I’m suggesting that more information and data are required to make the case here.

Validity of the findings

Based on the requirements of PeerJ, I have to state that the data do not appear robust, statistically sound, and controlled. Granted, fossils are what they are, and this is not what I’m concerned about. My concern is again that much more data could conceivably be obtained from a larger sample of readily available turkey, chicken, and other common bird necks. You may only have 2 good sauropod necks, but how you constrain the soft tissue should go beyond a single turkey. Or maybe you did look at more than one turkey, but this isn’t clear from your paper. Whereas the data and formulae for how the intervertebral extension was calculated are provided, they still rest on what appears to be a paucity of intervertebral cartilage data. It may turn out that Taylor is correct (I suspect he is, at least so far as sauropods having longer, more flexible necks than supposed in earlier studies), but it would very helpful to see this borne out with more data.

Additional comments

Mike, I hope you take my comments and criticisms in the spirit of producing a strong paper. I think you have the kernel of something interesting here, but I suspect it needs to be fleshed out more. My processor also autocorrected from UK English to USA English. If that doesn't matter, just ignore.

·

Basic reporting

Basics are fine

Experimental design

is fine.
I wish we could avoid the handwaving about the triangle being almost right-angled, but as practically nobody ever measures the necessary distances, and everyone measures what Taylor uses, I think it is best to simply live with the tiny inaccuracy.

Validity of the findings

The validity of the findings is limited by an omission in a previous publication that this MS is an add-on and follow-up to (Taylor and Wedel 2013), which can't be helped.

Additional comments

I remarked on various issues in the MS; nothing major.

In sum, this should have been in Taylor and Wedel 2013; it is worth publishing without much fuss.

·

Basic reporting

The manuscript is open about methods, with clear results in the figures and tables. The author covers the relevant literature well, tying his own past work to the current quantified treatment. See below for an additional reference.
The manuscript's structure would benefit from minor rearrangement and integration of sentences. Currently the Results have a combination of methods, results, an important new concept (cartilaginous neutral pose: CNP), and interpretation. The results point to the figures and tables for the final neck pose, but do not report their angular findings. The Discussion, conversely, has numerical results that are better suited to the Results. In addition to reporting quantitative posture results in the Results, I suggest putting the CNP in the introduction, as what your results will reveal.

Experimental design

The manuscript introduces an excellent method for approximating intervertebral angles. You may get criticisms for using radians and not sines of angles, but radians are clever and valid approximations at these angles. The method cries out for one equation, in addition to the sentence explaining the equation.
It may be worth addressing, in the methods or into, how much of the added cartilage is on the condyle and how much is in the cotyle. It looks like it's all condyle in the figures, and appears that way in Taylor and Wedel (2013). Sorry if I'm not reading things correctly. It seems that the results would be the same if there were equal amounts of cartilage on both ends, and there would be half the challenge of maintaining the avascular tissue.

Validity of the findings

The findings and conclusions may be even more valid than the author argues. Samman (2013) includes a Dinomorph T. rex model with a vertical ONP, and has interesting radiographs and dissections showing bird neck posture.
Samman, T. 2013. Tyrannosaurid craniocervical mobility: a preliminary assessment, pp. 195–210. In: Parrish J.M. et al., Eds. Tyrannosaurid paleobiology. Bloomington: Indiana University Press.
Strict ONP by various criteria can straighten the neck too much, either horizontally or vertically. In tyrannosaurids the shape of the centra dictate curvature better than the zygapophyses. Samman (2013) thought the T. rex model was the problem, but it looks more like Dinomorph was doing its job with a good zygapophyseal ONP.
I suggest giving Samman's work a shout-out as another objective example of how neutral postures are unrealistic. With T. rex a 500+ kg head dragged the neck (and extended its dorsal ligaments) into an S-curve. Diplodocids (like birds) had a pretty long neck past the inflection point to cause the same curvature.

Additional comments

It's amazing how the "strangely" curved diplodocid neck postures look like flensed bird necks.
Not to dictate style too much, but you can be a little more generous to yourself and colleagues' work in the introduction. You can make your case without any negative words towards your earlier oversights, or even towards Stevens and Parrish (see suggested wording in the commented manuscript). We'll all look better in 100 years if we're generous and use evidence to refine and correct past work (including Taylor and Wedel 2013) rather than emphasizing its flaws. I'm spectacularly hypocritical about this in draft replies to manuscript reviews, with lots of savage writing I delete later. Then I can go Jesus on them while obliterating their case with evidence. You win a fight you never have to start.
Again, try to maintain your style and personality, if you go along with the suggested high-mindedness.

---

## Round 0.2 · accepted · Accept

I am pleased with the changes made to the manuscript and find your rebuttals acceptable. The paper reads much more clearly now. I am happy to accept it for publication.